# The Role of Dietary Patterns and Dietary Quality on Body Composition of Adolescents in Chinese College

**DOI:** 10.3390/nu14214544

**Published:** 2022-10-28

**Authors:** Hongrui Li, Dajun Li, Xianyun Wang, Huini Ding, Qinghua Wu, Haojun Li, Xuan Wang, Kaifeng Li, Rong Xiao, Kang Yu, Yuandi Xi

**Affiliations:** 1Beijing Key Laboratory of Environmental Toxicology, School of Public Health, Capital Medical University, Beijing 100069, China; 2Research Institute, Heilongjiang Feihe Dairy Co., Ltd. C-16, 10A Jiuxianqiao Rd., Chaoyang, Beijing 100015, China; 3Department of Clinical Nutrition, Peking Union Medical College Hospital, Chinese Academy of Medical Sciences and Peking Union Medical College, Beijing 100730, China

**Keywords:** dietary pattern, diet balance index, dietary inflammatory index, bioelectrical impedance analyzer, body mass index, fat mass index, fat-free mass index, basal metabolic rate

## Abstract

There is limited evidence regarding the effects of dietary pattern and dietary quality on the risk of unhealthy weight status and related body composition in Chinese adolescence. In particular, studies using bioelectrical impedance analyzer (BIA) in these subjects are rare. The aim of this study was to evaluate the role of diet in body composition, to find a healthy dietary pattern for Chinese youth, and to promote the application of BIA among this population. A total of 498 participants aged from 18 to 22 years old were included. Dietary patterns were identified by principal components analysis. Energy-adjusted dietary inflammatory index (DII) and diet balance index (DBI) were calculated based on semi-quantitative food frequency questionnaire. Multivariate linear regression and logistic regression analysis were used to examine the relationship of dietary patterns, dietary quality with body mass index (BMI), fat mass index (FMI), fat-free mass index (FFMI), and the effect of dietary factors on BMI levels. The majority of participants with overweight and obesity had abdominal obesity, and there was 3.7% abdominal obesity in normal BMI individuals. Four dietary patterns were detected in the subjects. The pattern with the higher energy intake, which was close to the Western diet, was positively correlated with BMI (β = 0.326, *p* = 0.018) and FMI (β = 0.201, *p* = 0.043), while being negatively correlated with FFMI (β = −0.183, *p* = 0.021). Individuals who followed the pattern similar to the Mediterranean diet had a higher basal metabolic rate (BMR), and the highest fat free mass, soft lean mass, and skeletal muscle mass (*p* < 0.05) but the lowest FMI, visceral fat area (VFA), waist–hip ratio, and FMI/FFMI ratio (*p* < 0.05). Higher energy-adjusted DII was associated with high BMI. Higher bound score (HBS) (β = −0.018, *p* = 0.010) and diet quality distance (DQD) (β = −0.012, *p* = 0.015) were both negatively correlated with FFMI. In conclusion, fat or muscle indexes, such as BMR, FMI, and FFMI, had an important role in predicting overweight and obesity, which suggested the importance of applying BIA among Chinese college students. Students who followed healthful dietary patterns or the high-quality diet that is similar to the Mediterranean diet but not close to the Western diet were more likely to have a healthy BMI and normal body composition.

## 1. Introduction

Nowadays, the epidemic of obesity is one of the most severe health problems facing the world [1]. The prevalence of overweight and obesity has dramatically increased in the Chinese population, especially during adolescence [2,3]. Data shows 22.7% of males and 8.4% of females aged 19–22 years were overweight or obese among Chinese college students [3]. It is believed that one of the major causes of overweight and obesity in youth is poor dietary patterns [4]. College students undergo great changes during their years at school in terms of both physical and social development, resulting in being particularly vulnerable to nutritional disturbances. Additionally, college students are less likely to maintain a healthy lifestyle due to many factors, such as stress, lack of time, and changes in living arrangements [5]. Therefore, inappropriate dietary patterns and dietary quality might be the critical problem influencing the physical health of youth, such as body composition of fat, muscle, body water, basal metabolic rate (BMR), and so on.

Diets with a high intake of whole grain, vegetables, fruits, low and non-fat dairy, and lean meat are generally considered healthy dietary patterns. These patterns can reduce the risk of obesity, cardiovascular disease, and some cancers [6]. The Mediterranean diet has been identified as having proved to be the most effective pattern in terms of the prevention of obesity-related diseases [6]. The Dietary Approaches to Stop Hypertension (DASH) diet is also a good choice for weight management, particularly for weight reduction in participants with overweight or obesity [7]. Furthermore, a healthy dietary pattern is positively associated with high levels of cardiorespiratory fitness (CRF), which refers to the capacity of the circulatory and respiratory systems to supply oxygen to skeletal muscle mitochondria for the energy production needed during physical activity [8]. In that, the identification of dietary patterns for a specific population may be helpful in exploring their risk of diet-related diseases. The effects of dietary patterns on overweight/obesity have been widely investigated and confirmed in children, working people, and the elderly. However, they are not well-described in the distinctive group of adolescents in Chinese college [8]. The high intake of confectionery, fast foods, and beverages with sugar and the low consumption of fruit and vegetables are high-frequency events in Chinese college students [4,9]. Therefore, what dietary patterns are the characteristic of this group? How about their dietary quality? Whether dietary patterns directly contribute to the body composition anomalies? These questions need to be concerned for their health undoubtedly.

Changes in body composition occur when there is a mismatch between nutrient intake and requirement. Body composition analysis is useful in assessing the nutritional status of adolescence and monitoring the changes associated with growth and disease conditions [10,11]. The bioelectrical impedance analyzer (BIA) is a portable and time-efficient method, which is easily usable to estimate body composition parameters. This popular technology has shown high agreement with the gold-standard dual energy X-ray absorptiometry (DXA) [12,13] in the past few years. It has also been used to evaluate the nutritional status of some populations, including college students [14]. Fat mass index (FMI) and fat-free mass index (FFMI) indices eliminate the differences of body fat percentages associated with height, which could be useful for the measurement of related body composition [15]. Visceral fat area (VFA) is the measure of obesity (particularly abdominal obesity), which is a better indicator than fat mass [16,17]. However, BIA is less used in Chinese youths. In this study, these indicators and body mass index (BMI) were used comprehensively to evaluate the physical status of college students, and their correlation with nutrition was further detected.

The Diet Balance Index (DBI) is designed to assess undernutrition and overnutrition, based on the dietary guidelines for Chinese residents and the Pagoda of Balanced Diet published by the Chinese Nutrition Society. It has been widely used in nutritional research, so as to help people make decisions according to the dietary guidelines and assist with comprehensive evaluation for people with different dietary recommendations [16,18]. The Dietary Inflammatory Index (DII) was created in 2009 to quantify the effect of diet on inflammatory potential. Evidence shows that significant positive correlations were found between DII score and weight, BMI, waist-to-hip ratio, and body fat percentage [17]. Therefore, we used these two indices to evaluate the nutritional status of college students. The aim of this study was to explore the role of dietary patterns and dietary quality on the body composition and BMI of youth in Chinese college.

## 2. Materials and Methods

### 2.1. Participants

In this cross-sectional study, 498 college students (aged 18–22 years) who had taken a body composition survey and 488 who had taken a dietary intake survey in a university located in an urban area and its’ affiliated school located in a suburb in Beijing were assessed. The sampling of college students was carried out randomly according to different grades. Finally, 474 individuals finished all of the projects. All subjects signed informed consent before they were included.

### 2.2. Assessment of Dietary Intake

The Food Frequency Questionnaire (FFQ) of 2002 China National Nutrition and Health Survey (CNHS 2002) [19] was used to collect dietary intake information, which obtained the habitual consumption of foods over the past year. The total dietary intake of each nutrient was calculated by summing the nutrients from all food items reported in the FFQs.

### 2.3. Derivation of Dietary Patterns

Factor analysis of principal components (PCA) is a multivariate statistical procedure to empirically derive dietary patterns. After measuring the sampling adequacy by the Kaiser Meyer–Olkin test (0.749) and Bartlett’s test of sphericity (*p* = 0.000), PCA was applied to derive patterns from the food groups and determine factor loadings for each group. The varimax orthogonal rotation was carried out in order to simplify the interpretation of the data. The salt and soy sauce in 28 kinds of food were combined and named as “equivalent salt”. A total of 27 factors were retained based on the scree plot (Appendix A) and the standard criterion of eigenvalues >1.5, which were defined as major dietary patterns [20,21,22]. Absolute factor loadings above 0.30 were considered to contribute significantly to the pattern. Consequently, a four-factor solution was found, with factor 1 explaining 11.190% of the variance, factor 2 explaining 9.458%, factor 3 explaining 9.407%, and factor 4 explaining 7.462%. Together, they explained 37.517% of the total variance. The factor scores for each pattern and for each individual was determined by summing the intake of each food group weighted by the factor loading [21].

### 2.4. DII Score and Dietary Intake Assessment

The dietary inflammatory index (DII) was used to assess the inflammatory potential of diet according to Shivappa et al. [23] and our previous study [24]. The energy-adjusted DII was calculated to reduce the bias induced by different energy intake in accordance with the published procedures [25,26].

The daily intake on average for each food item was calculated according to the Diet Balance Index 2016 (DBI-16) [16], a revised version from the Chinese Diet Balance Index 2007 (DBI-07) [27]. DBI-2016 comprises 14 subgroups of 8 components from the dietary guidelines for Chinese residents [28], including: (1) cereal; (2) vegetable and fruit; (3) dairy and soybean; (4) animal food (red meats/products/poultry/game, fish/shrimp, and egg); (5) empty energy foods (cooking oils and alcoholic beverage); (6) condiments (additional sugar, and salt); (7) diet variety; and (8) drinking water. A score of 0 for each DBI-16 component means that the individual has achieved the recommended intake of the corresponding food group. Positive scores (ranging from 1 to 12) indicate the excessive intake level of cereals, red meat/products/poultry/game, eggs, cooking oils, alcoholic beverage, addible sugar, and salt, while negative scores (ranging from −12 to −1) indicate the inadequate intake level of cereals, vegetables, fruits, dairy, soybeans, red meat/products/poultry/game, fish/shrimps, eggs, diet variety, and drinking water. Considering the difference of nutrient requirements in energy consumption, the scoring of these 14 food subgroups was based on 11 levels of energy intake.

Based on the scores for each DBI-16 component, three indicators of diet quality were calculated: (1) the lower bound score (LBS), an indicator for inadequate food intake, was computed by adding all the negative scores; (2) the higher bound score (HBS), an indicator for excessive food intake, was calculated by adding all the positive scores; (3) the diet quality distance (DQD), an indicator of unbalanced food intake, is calculated by adding the absolute values of both positive and negative scores [16]. The ranges of LBS, HBS, and DQD were 1 to 41, 1 to 31, and 10 to 61, respectively.

### 2.5. Body Composition Assessment

Body compositions were assessed using of a BIA (Inbody 770 Co., Seoul, Korea). Measurements were taken according to the manufacturer’s protocol [29]. An experienced tester instructed that each participant stood and held the handle accurately so that the voltage was always measured at the same point. For accuracy, please see the consistency verification study of BIA and DXA conducted at the Mayo Clinic [30]. Our participants were in a fasting condition and urinated just before the body composition analysis to obtain the exact result. This device directly estimated the raw bioimpedance parameters (e.g., impedance, reactance, and phase angle). The related equation could be found in the professional research [13]. BMI was calculated as the ratio of body mass to height squared (kg/m^2^). The calculation of other body components was obtained by Inbody 770 Co. through electrophysiology combined with the relevant literature (not publicly available).

According to the Working Group on Obesity in China, obesity was defined as a BMI of at least 28 kg/m^2^ and overweight was defined as 24 kg/m^2^ ≤ BMI < 28 kg/m^2^. A person with 18.5 kg/m^2^ ≤ BMI < 24 kg/m^2^ was normal weight, and <18.5 kg/m^2^ constituted thinness [21]. Body composition was assessed, including FMI, FFMI, and so on. The percentage body fat (PBF) indicates the significant percentage of body fat to body mass. The body weight without fat is “fat-free mass”. VFA was an assessment of fat stored in the abdominal cavity. A waist circumference above 90 cm for male and 85 cm for female is abdominal obesity [31].

### 2.6. Statistical Analysis

After testing the normality of data by the Shapiro–Wilk test or the Kolmogorov–Smirnova test, the variables were presented as means or medians, followed by the SD or interquartile range. One-way analysis of variance (ANOVA) with Bonferroni’s correction or Kruskal–Wallis tests were applied to compare means or medians across different BMI levels and the scores of dietary patterns. The χ2 test was used for categorical variables, which were presented using absolute and relative frequencies. To examine the associations of dietary patterns, diet quality, body composition, and BMI, both multivariate linear regression models and multiple logistic regression were employed. Specifically, Model 1 was unadjusted, while Model 2 was adjusted for gender, age, BMR, and total energy intake (kcal). All regression models were assessed for normality by the examination of the model residuals plotted. After evaluating collinearity and residuals, the results were presented containing the β coefficients or odds ratio (OR) and their respective 95% confidence intervals (CIs). The continuous independent variables were converted into categorical variables in order to use multivariate logistic regression. The classification principles were as follows: patterns 1–4 and DII were divided by quartile, and HBS, LBS, and DQD were divided into three grades by DBI-16 (HBS: 1–9 was suitable, 10–18 was low excessive intake, and over 19 was medium high excessive intake; LBS: 1–12 was suitable, 13–24 was low inadequacy intake, and above 25 was medium and high inadequacy intake. DQD: 1–17 is suitable, 18–34 is low dietary imbalance, and above 35 is moderate to high dietary imbalance). A series of body composition indicators (except FMI and FFMI [32]) were divided into two categories according to related standards. The references for body fat percentage, abdominal obesity, visceral fat area, and waist–hip ratio were <25% in male and <30% in female, male < 90 cm and female < 85 cm [33], below 100 cm^2^ and <0.9 in male and <0.85 in female [34]. *p* < 0.05, which was two-tailed, was considered statistically difference. All analyses were performed using the statistical software package IBM SPSS Statistics version 21^®^ (IBM Corp., Armonk, NY, USA).

## 3. Results

### 3.1. Overall Sample Characteristics

Participants were divided into four groups according to BMI (Table 1). There was no difference in age among four groups, but the gender composition was significantly different (*p* < 0.05). The rates of overweight/obesity were much higher in male than in female. There was a statistical difference in abdominal obesity among the four groups (*p* < 0.05). No participants in the thinness group suffered from abdominal obesity, while 96.3% subjects with obesity had abdominal obesity. Interestingly, there was 3.7% abdominal obesity in normal BMI individuals, and almost half of subjects had abdominal obesity in overweight college students. Compared with the normal-weight group, all of the indices related to muscle mass and fat mass in the overweight and obesity groups were higher than those in the normal-weight group, but the indices in the thinness group had completely opposite results (*p* < 0.05). Fat-free mass, soft lean mass, skeletal muscle mass, and skeletal muscle mass index of the individuals in the obesity group were significantly higher than those in the overweight group (*p* < 0.05). People with obesity had more intake of flour and fried foods. Students with overweight and obesity consumed more energy than normal-weight students (*p* = 0.062), which could be seen as demonstrating a statistical difference. In addition, the energy-adjusted DII score had a higher trend in participants with obesity compared with the subjects in other groups, but no significant difference could be found. There was no significant difference in DBI-16 score as well.

### 3.2. Dietary Pattern Characterization

Four dietary patterns were identified and labeled: pattern 1, pattern 2, pattern 3, and pattern 4. These patterns explained 11.190%, 9.458%, 9.407%, and 7.462% of the total variance, respectively. Pattern 1 was characterized by aquatic products, nuts, processed meat, bean productions, and beer. Pattern 2 was characterized by cereals, tubers, fruits, flour food, and milk. Pattern 3 was characterized by high-calorie foods and the high intake of salt, including fried food, sweets and desserts, red meat, equivalent salt, flour food, poultry, oil, and drinks with sugar and rice. Pattern 4 was characterized by the intake of various vegetables, mushroom, red meat, red wine, and eggs (Table 2).

The distribution of the four dietary patterns of college students was different in gender. The distribution of girls tends to be even in four patterns. On the contrary, boys were more inclined to consumption pattern 3 (29.9%) and pattern 4 (36.1%). Students who followed pattern 3 had the highest FMI, VFA, waist–hip ratio, and FMI/FFMI ratio compared with the people following other patterns (*p* < 0.05). Individuals who followed pattern 4 had the lowest FMI, VFA, waist–hip ratio, and FMI/FFMI ratio but had the highest FFM, soft lean mass, and skeletal muscle mass (*p* < 0.05) (Table 3).

The nutrient characteristics of four patterns were also evaluated in this study. Participants following pattern 1 had a higher percentage of energy intake from protein and more bean-derived protein. A higher intake of PUFA, including DHA and EPA, was also the characteristic of pattern 1 (*p* < 0.05). Participants following pattern 2 had a higher percentage of energy intake from carbohydrates, and the intake of both vitamin C and potassium were higher (*p* < 0.05). Pattern 3 had the highest intake of fat and a higher percentage of energy intake from carbohydrates. Subjects following pattern 3 had a higher intake of vitamin B_3_ (*p* < 0.05). The feature of pattern 4 was a higher percentage of energy intake from protein and a higher intake of animal-derived fat and PUFA (*p* < 0.05). Moreover, energy-adjusted DII was higher in pattern 2, and the DBI-16 score tended to be higher in pattern 3 (*p* < 0.05) (Table 3).

### 3.3. Association of Dietary Patterns and Dietary Scores with BMI

Linear regression analysis showed that FMI and FFMI were positively correlated with BMI in all three models (*p* < 0.05). Model 1 is adjusted by no factors. Model 2 is adjusted for age and gender. Model 3 is additionally adjusted for BMR and energy based on model 2. In addition, pattern 2 was positively correlated with BMI in model 2 (β 0.339, 95% CI 0.068, 0.610, *p* = 0.014), but it disappeared when in model 3 (β −0.032, 95% CI −0.478, 0.413, *p* = 0.886). A similar result could be found in pattern 3 as well (model 1: β 0.326, 95% CI 0.056, 0.595, *p* = 0.018; model 3: β −0.151, 95% CI −0.603, 0.301, *p* = 0.512). Energy-adjusted DII was positively associated with BMI in model 1 (β 0.065, 95% CI 0.022, 0.107, *p* = 0.003) and model 2 (β 0.046, 95% CI 0.002, 0.089, *p* = 0.042), but it disappeared in model 3 (Table 4).

Multiple logistic regression showed that high FMI (OR 15.90, 95% CI 7.76, 32.59, *p* = 0.000) and FFMI (OR 5.91, 95% CI 3.65, 9.58, *p* = 0.000) were the risk factors for overweight, and high FFMI was a risk factor for obesity (OR 15.29, 95% CI 6.92, 33.77, *p* = 0.000) as well. In addition, abdominal obesity (OR 9.77, 95% CI 2.86, 33.34, *p* = 0.000), high level of VFA (OR 6.83, 95% CI 2.54, 18.40, *p* = 0.000) and a high level of PBF (OR 18.69, 95% CI 3.71, 94.25, *p* = 0.000) were risk factors for overweight, while waist–hip ratio (OR 0.25, 95% CI 0.09, 0.67, *p* = 0.006) was a protective factor. No risk of pattern 2 and pattern 3 was found in overweight and obesity. Interestingly, Q2 of pattern 4 (OR 0.04, 95% CI 0.00, 0.95, *p* = 0.047) was a protective factor for obesity. Q2 of energy-adjusted DII (OR 2.26, 95% CI 0.96, 5.31, *p* = 0.062) could be considered a risk factor for overweight. In addition, moderate and high levels of LBS (OR 8.04, 95%CI 1.28, 50.51, *p* = 0.026) was a risk factor for overweight (Table 5).

FMI (OR 0.07, 95% CI 0.03, 0.18, *p* = 0.000), FFMI (OR 0.08, 95% CI 0.03, 0.20, *p* = 0.000), and PBF (OR 0.05, 95% CI 0.01, 0.26, *p* = 0.000) were protective factors for thinness. Q3 of pattern 2 (OR 0.24, 95% CI 0.05, 1.13, *p* = 0.071) could be seen as a protective factor for thinness. Medium- and high-level LBS (OR 26.62, 95% CI 2.03, 349.71, *p* = 0.013) was a risk factor for thinness, while low- (OR 0.07, 95% CI 0.01, 0.49, *p* = 0.008), medium-, and high-level DQD (OR 0.03, 95%CI 0.00, 0.40, *p* = 0.008) were protective factors for thinness (Table 5).

### 3.4. Association of Dietary Patterns and Dietary Scores with Body Compositions

Pattern 1 had no significant correlation with FMI and FFMI in all three models (*p >* 0.05) (Table 6). Participants with a higher pattern 2 score had higher FMI in model 1 and higher FFMI in model 2, but the relations all disappeared in model 3. Participants who adhered pattern 3 had a higher FMI in model 2 (β 0.201, 95% CI 0.006, 0.395, *p* = 0.043). For the relationship between pattern 3 and FFMI, the regression coefficient was changed from positive to negative after adjusting model 3 (β −0.183, 95% CI −0.338, −0.027, *p* = 0.021). Pattern 4 had a negative correlation with FMI (β −0.225, 95% CI −0.427, 0.023, *p* = 0.029) but a positive correlation with FFMI (β 0.384, 95% CI 0.217, 0.552, *p* = 0.000) in model 1. Participants with high HBS had high FFMI in model 1 (β 0.036, 95% CI 0.005, 0.067, *p* = 0.022), which changed into a negative correlation after adjusting model 3 (β −0.018, 95% CI −0.031, −0.004, *p* = 0.010). HBS was also negatively correlated with FMI in all three models (*p* < 0.05). LBS was negatively correlated with FFMI in model 2 (β −0.031, 95% CI −0.058, −0.005, *p* = 0.022). Furthermore, DQD was negatively correlated with FFMI both in models 2 and 3 (β −0.021, 95% CI −0.036, −0.006, *p* = 0.007; β −0.012, 95% CI −0.022, 0.002, *p* = 0.015). The negative correlation was also found in DQD and FMI in model 1, but it disappeared after adjusting model 2 and 3 separately. Energy-adjusted DII was positively correlated with FMI for *p* = 0.061 in model 2 (β 0.024, 95% CI −0.011, 0.037), and it was positively correlated with FFMI in both models 1 and 2 (β 0.041, 95% CI 0.018, 0.063, *p* = 0.000; β 0.022, 95% CI 0.007, 0.038, *p* = 0.005).

Participants following pattern 3 had higher red meat products and poultry intake of HBS (*p* < 0.05), while the variation in LBS was insignificant among different dietary patterns. Red-meat-derived protein was higher in pattern 3 and pattern 4. In pattern 3, poultry-derived protein was higher. Pattern 1 and pattern 3 were higher than pattern 2 and pattern 4, respectively, in processed-meat-derived protein (*p* < 0.05). Participants following pattern 4 had a higher egg intake of LBS, which was similar to the RNIs/AIs of eggs than pattern 1 and pattern 3 (*p* < 0.05) while HBS was insignificant between different dietary patterns. (Appendix A).

## 4. Discussion

The abnormal body composition of youth college students seriously affected their physical health, and further increased the risk of long-term chronic diseases, such as hypertension, diabetes, and cardiovascular diseases [35]. In our subjects, 17.06% had overweight, and 5.42% had obesity. The rates of overweight/obesity were much higher in male than in female. Regarding the problem of abdominal obesity in college students, it needs to be noticed that, not only people with overweight and obesity accounted for a high proportion (49.4% and 96.3%, respectively), but also 3.7% of abdominal obesity was invisible in a person of normal BMI. Therefore, BMI could not fully reflect the physical health of the youth, and body composition should be considered comprehensively. This opinion has also been proven by others [8], that FMI and FFMI could be useful for the measurement of obesity in college students. Our results showed that fat-related indices, such as PBF, FMI, and VFA, and the waist–hip ratio and FMI/FFMI were much higher in people with overweight/obesity. In addition, the fat-free indices including FFM, FFMI, soft lean mass, skeletal muscle mass, and skeletal muscle index were also higher in a person with overweight/obesity. This phenomenon was similar to the research [36] that indicated people with high BMI exhibited higher total leanness on the Mediterranean diet.

Four dietary patterns calculated by a factor analysis of principal components were discovered in our subjects, which were found to be correlated with BMI and body composition.

Pattern 3, which we named ‘Western dietary pattern’, was positively associated with BMI in this study. Because of the highest intake of fat compared with other dietary patterns, the fat-related indices including FMI, VFA, and waist–hip ratio were the highest in youths who preferred pattern 3 than others, but the fat-free indices, such as FFMI and skeletal muscle index, were not. These were similar to the other studies [37,38], which showed that the odds ratios of obesity increased across the quartiles of the nutritional pattern primarily consisting by saturated/mono-unsaturated fatty acids. A recent study found [39] that the animal-driven nutrient pattern (characterized by animal protein and saturated fat) was significantly associated with the increase of FMI, which is similar to our result. Another characteristic of pattern 3 was sweets and desserts. There is similar evidence that unhealthy plant foods, including fruit juices and sugar-sweetened beverages, could increase the risk of abdominal obesity [39], and the consumption of sugars seems to have a close relationship with the variables of body composition, especially in visceral fat and muscle mass [40]. Furthermore, there is a significant correlation between the consumption of the vitamin B group (vitamin B_3_, with OR = 1.01) and an increased risk of excess weight in those aged 6–18 years [40], and the plasma concentrations of vitamin B_3_ in the Emirati population with obesity were significantly higher than those in healthy volunteers [41]. A similar result was found in our study. More interestingly, we found that, after adjusting for BMR and energy, the original no association between pattern 3 and FFMI turned into a significantly negative correlation. In this case, the energy intake and BMR of individuals should be taken into account in evaluating the physical health of youth. As the above results showed, the higher incidence of overweight and obesity in young male college students might be related to a stronger tendency towards the unhealthy dietary habits of pattern 3, especially abdominal obesity.

The Mediterranean diet represents a dietary pattern that incorporates the healthy traditional eating habits of populations from countries surrounding the Mediterranean Sea, which includes the high consumption of vegetables, fruit, legumes, nuts, beans, fish, and unsaturated fats, such as olive oil and red wine [42,43]. In the present study, pattern 4, which was partly consistent with the Mediterranean diet, provided a similar conclusion, that vegetable-based nutrient patterns have been linked to a lower risk of overweight and obesity [44]. More evidence showed that numerous vegetables were inversely related to weight and the consumption of fruits and vegetables was related to a low percentage of fat [29]. Nutrients analyses found that, not only the consumption of lower carbohydrate and total fat, but the proper number and quality of protein (more bean-derived protein and not less animal food-derived protein) is another characteristic of pattern 4. This was definitely an important reason for good health, with effective body fat control in college students, which could find similar results in research of a low-carbohydrate Mediterranean (LCM) diet [45]. The proper proportion of red meat (Table 2 and Appendix A) was also the healthy beneficial factor of pattern 4, which may also be reflected in the negative correlation between HBS and FMI (Table 3 and Table 6). Students who followed pattern 4 had a lower VFA and FMI, while some muscle-related indicators such as FFMI, SLM, FFM, and SMM tended to be higher than other patterns. This was consistent with the results that a high level of adherence to the Mediterranean diet was associated with a high level of muscular fitness in Spanish university students [36]. More proof [46] demonstrated that women with higher adherence to the Mediterranean diet had higher appendicular lean mass index (ALMI, appendicular lean mass/height squared).

In order to further exploring the quality of different diet, DBI-16 and energy-adjusted DII was calculated [16,23,24] and used to describe dietary patterns in the present study. Results that supported pattern 2 had higher DII, and pattern 3 also tended to be the unbalanced diet. This was consistent with the previous results of overweight/obesity and body fat distribution. The same conclusion could be found in other research [47,48,49], which showed that high dietary quality in adolescents can improve nutrient status and be associated with a lower percentage of body mass. Samanta T. Valdés [45] reported that overweight individuals were more predisposed to vitamin deficiency, due to differences in the intake of dietary fruit, vegetables, and energy by individuals with overweight and obesity as compared to individuals of normal weight.

Linear regression analysis and multiple logistic regression showed more interesting and valuable results. Results confirmed the FMI and FFMI could be used for the measurement of obesity in college students. Second, based on different dietary habits, BMR and total energy intake could be the main determinants for the physical health based on different dietary habits among young people. Moreover, dietary quality is significantly associated with BMI. DII, which was adjusted by energy, was positively correlated with BMI. A moderate and high level of LBS was the risk factor for both thinness and overweight. The valuable explanation could be found in Table 6. Not only BMI, but also FMI and FFMI (both fat and muscle indices), need to be concerned comprehensively in the evaluation of relationships.

In addition, some of the diet-related indices mentioned in this paper, such as DII, have been combined with BIA applications [50]. A recent paper [29] used the BIA technique (InBody 770 scanner), which is highly in accordance with our study, to find the relationship between nutrient patterns and metabolic syndrome. Several obesity-related studies were also using the BIA device [29,50,51,52,53]. These indicated that BIA can be well developed in the most areas. Therefore, it is necessary to pay attention to the promotion of BIA among college students, especially Chinese youth.

There were two key points showed in the present study worthy of being paid attention to in the young population. One was that BIA is the best option because of the better consistency with DXA. Although the application of this method is still widely used in people with obesity [29,50,51,52,53], the related study for Chinese college students [54] is limited. Therefore, BIA may be applied in Chinese college students with abnormal BMI or body composition in the future. Another concern is normal-weight obesity (NWO). BMI is usually used to evaluate overweight/obesity. However, 13 of the 352 normal BMI students in present study had abdominal obesity. The incidence of normal-weight obesity (NWO) need to be concerned. BIA-derived indicators, such as FMI and FFMI, BMR and energy intake, should be used comprehensively to analyze the physical health of college students.

There were some limitations as follows. Firstly, it has been reported that the assessment of nutrient patterns rather than dietary patterns has more advantages resulting from the structures. Their structures are not affected by special traditional behavior such as food preparation. Secondly, no causal inference could be made due to the cross-sectional nature of the study. At last, the limited number of participants might reduce the representativeness of objective population. Further studies should be conducted in future.

## 5. Conclusions

BIA-derived indexes, such as BMR, FMI, and FFMI, might be key determinants in the definition of overweight and obesity in the youth population. A high-calorie dietary pattern that are highly loaded with refined grains, processed food, and condiments, which was close to the Western diet, could be the main dietary risk factor for obesity and abnormal fat distribution. A healthful dietary pattern, characterized by the consumption of various vegetables, red wine, and a high quality of protein, similar to the Mediterranean diet, could help maintaining a reasonable weight and BMR in youth. Energy-adjusted DII and DBI-16 were could be used to effectively assess dietary quality and the risk of overweight/obesity. The application of BIA in Chinese college students should be emphasized. These findings provide new perspectives for future health promotion interventions and behavior change among Chinese youth.

## Figures and Tables

**Table 1 nutrients-14-04544-t001:** Overall sample characteristics stratified by BMI.

Variable	Total	Thinness	Normal Weight	Overweight	Obesity	*p*
**General**						
**N (%)**	498	34 (6.82)	352 (70.68)	85 (17.06)	27 (5.42)	
Age, year	20 (19,22)	20 (19,22)	20 (19,22)	20 (19,23)	20 (19,22)	0.792
Male, *n* (%)	113 (22.7)	6 (5.3)	66 (58.4) ^e^	26 (23.0)	15 (13.3) ^e^	0.000 **
Female, *n* (%)	385 (77.3)	28 (7.3)	286 (74.3) ^e^	59 (15.3)	12 (3.1) ^e^	0.000 **
**Body comp** **osition**						
**N**	498	34	352	85	27	
BMR, kcal	1238.0 (1164.5, 1383.3)	1135.5 (1099.5, 1210.8)	1224.0 (1162.0, 1302.5)	1352.0 (1240.5, 1565.5)	1674.0 (1383.0, 1787.0)	0.000 **
Abdominal obesity						
Yes	81 (16.3)	0 (0.0)	13 (3.7) ^c,d^	42 (49.4) ^b^	26 (96.3) ^b^	0.000 **
No	417 (83.7)	34 (100.0)	339 (96.3)	43 (50.6)	1 (3.7)	
PBF, %	30.2 (24.4, 34.3)	23.1 (21.3, 27.4) ^b–d^	29.6 (24.4, 33.7) ^a,c,d^	33.7 (27.7, 37.5) ^a,b^	34.3 (28.8, 41.1) ^a,b^	0.000 **
FMI	6.4 (5.0, 8.1)	4.3 (3.7, 4.9) ^b–d^	6.1 (5.0, 7.5) ^a,c,d^	8.7 (7.1, 9.5) ^a,b^	10.2 (8.7, 12.8) ^a,b^	0.000 **
VFA (cm^2^)	73.2 (55.4, 101.4)	42.8 (37.9, 56.1) ^b–d^	69.3 (54.3, 91.2) ^a,c,d^	110.3 (83.5, 122.1) ^a,b^	140.2 (103.3, 175.8) ^a,b^	0.000 **
Waist–hip ratio	0.84 (0.81, 0.87)	0.80 (0.78, 0.82) ^b–d^	0.83 (0.81, 0.86) ^a,c,d^	0.86 (0.84, 0.91) ^a,b^	0.92 (0.88, 0.94) ^a,b^	0.000 **
FFM, kg	40.2 (36.8, 46.9)	35.5 (33.8, 38.9) ^b–d^	39.5 (36.7,43.2) ^a,c,d^	45.5 (40.3, 55.4) ^a,b,d^	60.3 (46.9, 65.6) ^a–c^	0.000 **
FFMI	15.1 (14.3, 16.8)	13.5 (13.1, 13.1) ^b–d^	14.9 (14.2, 15.9) ^a,c,d^	16.9 (15.8, 18.8) ^a,b^	19.6 (17.3, 20.8) ^a,b^	0.000 **
Soft lean mass, kg	37.7 (34.5, 44.1)	33.3 (31.6, 36.5) ^b–d^	37.0 (34.4, 40.5) ^a,c,d^	42.7 (37.9, 52.1) ^a,b,d^	56.9 (44.1, 61.7) ^a–c^	0.000 **
Skeletal muscle mass, kg	21.6 (19.6, 25.6)	18.9 (17.7, 20.9) ^b–d^	21.3 (19.6, 23.4) ^a,c,d^	24.7 (21.8, 31.0) ^a,b,d^	33.8 (25.8, 36.9) ^a–c^	0.000 **
Skeletal muscle index, kg/m^2^	6.2 (5.7, 7.0)	5.4 (5.2, 5.8) ^b–d^	6.0 (5.7, 6.5) ^a,c,d^	6.9 (6.4, 7.7) ^a,b,d^	8.2 (7.3, 8.9) ^a–c^	0.000 **
FMI/FFMI ratio	0.43 (0.32, 0.52)	0.30 (0.27, 0.38) ^b–d^	0.42 (0.32, 0.51) ^a,c,d^	0.51 (0.38, 0.60) ^a,b^	0.52 (0.40, 0.70) ^a,b^	0.000 **
**Food**						
**N**	474	33	336	80	25	
Flour, g/d	25.0 (10.5, 50.0)	20.5 (4.5, 50.0) ^b,d^	32.0 (10.5, 50.0) ^a^	25.0 (20.5, 50.0)	50 (20.5, 76.5) ^a^	0.017 *
Fried food, g/d	10.5 (3.5, 21.0)	3.5 (1.5, 15.8)	10.5 (3.5, 21.0)	10.5 (3.5, 29.9)	21.0 (7.0, 28.2)	0.036 *
**Energy**						
**N**	474	33	336	80	25	
Daily energy intake, kcal	1761.9 (1424.2, 2261.1)	1596.2 (1427.2, 1892.0)	1746.7 (1397.2, 2256.9)	1853.8 (1497.4, 2390.8)	1981.3 (1429.4, 2742.9)	0.062
**Dietary scores**						
**N**	474	33	336	80	25	
DBI-16						
HBS	12 (9, 16)	13 (10, 16)	12 (9, 16)	11 (9, 15)	12 (8, 17)	0.671
LBS	16 (14, 19)	17 (15, 19)	16 (13, 19)	16 (14, 18)	15 (12, 19)	0.712
DQD	28 (24, 34)	30 (26, 34)	28 (24, 35)	28 (24, 31)	27 (23, 33)	0.548
DII	6.7 (0.8, 10.4)	5.5 (1.3, 7.6)	6.7 (0.3, 10.4)	6.6 (3.5, 11.0)	8.0 (2.7, 12.4)	0.157

Abbreviation: BMR: basal metabolic rate; PBF: percentage body fat; FMI: fat mass index; VFA: visceral fat area; FFM: fat-free mass; FFMI: fat-free mass index; DBI-16: diet balance index-16; HBS: high bound score; LBS: low bound score; DQD: diet quality distance; DII: dietary inflammatory index. * *p* < 0.05, ** *p* < 0.01. For the DBI-16, The closer the score is to 0, the closer the food intake is to the recommended intake. For the DII, higher scores are more pro-inflammatory and lower scores are anti-inflammatory. ^a^
*p* < 0.05, statistically different with thinness; ^b^
*p* < 0.05, statistically different with normal weight; ^c^
*p* < 0.05, statistically different with overweight; ^d^
*p* < 0.05, statistically different with obesity. ^e^
*p* < 0.05, statistical difference between male and female.

**Table 2 nutrients-14-04544-t002:** Factor loading matrix for the four major dietary patterns identified from the Food Frequency Questionnaire (FFQ).

Dietary Patterns (*n* = 488)				
	Pattern 1	Pattern 2	Pattern 3	Pattern 4
**Variance explained (%)**	11.190	9.458	9.407	7.462
Food and food groups	Factor loadings			
River food	**0.801**	0.126	0.003	−0.024
Seafood	**0.754**	0.224	−0.137	0.008
Beer	**0.722**	−0.155	0.195	−0.072
Nuts	**0.547**	**0.356**	−0.108	0.153
Processed meat	**0.544**	−0.091	**0.416**	−0.043
Bean products	**0.470**	0.168	0.054	**0.362**
Congee	0.234	0.029	0.02	0.005
Tubers	0.143	**0.593**	0.163	0.058
Coarse cereal	0.247	**0.585**	−0.022	0.130
Fruits	0.100	**0.573**	−0.093	−0.039
Milk	−0.030	**0.560**	0.195	−0.088
Flour food	0.031	**0.419**	**0.497**	−0.069
Fried food	0.131	−0.078	**0.621**	0.199
Sweets and desserts	−0.001	0.269	**0.537**	−0.067
Equivalent salt	−0.070	−0.059	**0.497**	−0.120
Poultry	0.294	0.166	**0.440**	0.251
Oil	−0.064	0.038	**0.438**	−0.212
Drink with sugar	0.065	−0.208	**0.432**	0.278
Rice	−0.005	**−0.304**	**0.325**	0.208
Red meat	0.089	0.146	**0.537**	**0.313**
Dumpling	0.072	0.252	0.265	0.227
Egg	0.011	0.036	0.049	**0.604**
Light green vegetables	0.134	**0.387**	0.016	**0.588**
Dark green vegetables	0.083	**0.439**	−0.100	**0.550**
Mushroom	**0.407**	0.278	−0.046	**0.410**
Red wine	0.014	−0.157	0.003	**0.337**
Chinese liquor	−0.069	−0.099	0.042	0.296

Factor loadings with absolute values <0.30 are in bold.

**Table 3 nutrients-14-04544-t003:** Overall sample characteristics stratified by dietary patterns.

Variable	Pattern 1	Pattern 2	Pattern 3	Pattern 4	*p*
**General**					
**N**	122	122	122	122	
Age, year	20 (19, 23)	20 (19, 21)	20 (19, 21)	20 (19, 22)	0.477
Male, *n* (%)	31 (24.4)	14 (11.0) ^e^	38 (29.9)	44 (36.1) ^e^	0.000 **
Female, *n* (%)	91 (25.2)	108 (29.9) ^e^	84 (23.3)	78 (21.6) ^e^	
**Body composition**					
**N**	120	118	120	117	
BMI, kg/m^2^	21.7 (20.2, 24.1)	22.1 (20.4, 24.4)	22.6 (21.0, 24.4)	21.7 (20.1, 23.8)	0.190
Thinness	18.3 ± 0.1	18.0 ± 0.5	17.7 ± 0.7	17.6 ± 0.6	0.083
Normal weight	21.2 ± 1.4	21.3 ± 1.4	21.6 ± 1.4	21.3 ± 1.4	0.197
Overweight	25.2 (24.6, 26.0)	25.7 ± 1.1	25.4 ± 1.1	25.2 (24.7, 25.9)	0.601
Obesity	30.5 ± 2.0	29.8 ± 1.5	30.5 ± 1.8	30.6 ± 2.0	0.835
BMR, kcal	1248.5 (1186.3, 1423.8)	1239.0 (1165.0, 1345.3)	1273.0 (1196.0, 1511.3)	1282.0 (1188.7, 1508.5)	0.034 *
Abdominal obesity					
Yes	101 (25.6)	104 (26.3)	99 (25.1)	91 (23.0)	0.241
No	18 (23.7)	14 (18.4)	19 (25.0)	25 (32.9)	
PBF, %	29.9 (24.3, 33.5)	31.6 (26.2, 35.7) ^d^	31.6 (25.2, 34.9)	28.0 (19.8, 33.8) ^b^	0.008 **
FMI	6.3 (5.1, 7.9)	6.9 (5.4, 8.5) ^d^	7.2 (5.2, 8.5)	6.1 (4.3, 8.1) ^b^	0.016 *
VFA (cm^2^)	71.3 (56.4, 98.6)	81.2 (59.5, 109.8) ^d^	83.9 (60.4, 111.6) ^d^	68.6 (48.0, 103.4) ^b,c^	0.010 *
Waist–hip ratio	0.83 (0.81, 0.86)	0.84 (0.81, 0.88)	0.85 (0.82, 0.89) ^d^	0.83 (0.80, 0.87) ^c^	0.026 *
FFM, kg	40.7 (37.8, 48.8)	40.3 (36.8, 45.1)	41.8 (38.2, 52.9)	42.2 (37.9, 52.7)	0.034 *
FFMI	15.3 (14.4, 17.1)	15.1 (14.3, 16.5)	15.3 (14.4, 17.9)	15.4 (14.4, 17.8)	0.150
Soft lean mass, kg	38.2 (35.5, 45.7)	37.9 (34.5, 42.3)	39.2 (35.8, 49.9)	39.7 (35.5, 49.7)	0.032 *
Skeletal muscle mass, kg	21.8 (20.1, 26.5)	21.5 (19.7, 24.8)	22.5 (20.5, 29.7)	23.0 (20.3, 29.6)	0.030 *
Skeletal muscle index, kg/m^2^	6.2(5.8, 7.2)	6.0 (5.8, 6.6)	6.3 (5.8, 7.7)	6.4 (5.8, 7.5)	0.058
FMI/FFMI ratio	0.43(0.31, 0.50)	0.46 (0.35, 0.56) ^d^	0.46 (0.33, 0.54)	0.39 (0.25, 0.51) ^b^	0.000 **
**Nutrients**					
**N**	122	122	122	122	
**Energy**					
Daily energy intake, kcal	2034.4 (1665.4, 2661.6) ^b,c^	2382.8 (1897.0, 2989.6) ^a^	2495.0 (2042.0, 2968.8) ^a,d^	2075.5 (1718.2, 2800.9) ^c^	0.000 **
**Macronutrients**					
Protein, g/d	86.5 (65.9, 106.2)	90.9 (72.0, 114.0)	91.6 (77.4, 114.0)	87.8 (69.1, 113.0)	0.152
Animal-food-derived (%)	0.48 ± 0.13	0.52 ± 0.14	0.53 ± 0.13	0.51 ± 0.13	0.006 **
Bean-derived (%)	0.17 (0.11,0.26) ^b,c^	0.13 (0.05, 0.20) ^a^	0.10 (0.05, 0.17) ^a,d^	0.15 (0.07, 0.25) ^c^	0.000 **
Fat, g/d	113.3 (92.4,133.6) ^c^	112.9 (98.7, 135.9)	126.3 (109.9, 143.4) ^a,d^	108.4 (93.8, 134.4) ^c^	0.000 **
Animal-food-derived (%)	0.32 ± 0.11 ^d^	0.32 ± 0.11 ^d^	0.36 ± 0.10	0.36 ± 0.10 ^a,b^	0.003 **
Bean-derived (%)	0.09 (0.06, 0.13) ^b,c^	0.06 (0.03, 0.10) ^a^	0.05 (0.03, 0.09) ^a,d^	0.09 (0.04, 0.14) ^c^	0.000 **
Carbohydrate, g/d	198.4 (140.2, 253.6) ^b,c^	255.3 (187.7, 330.0) ^a,d^	241.9 (192.9, 319.9) ^a,d^	190.6 (140.35, 298.7) ^b,c^	0.000 **
Animal-food-derived (%)	0.17(0.09, 0.27)	0.19 (0.09, 0.3)	0.16 (0.1, 0.23)	0.14 (0.07, 0.23)	0.033 **
Bean-derived (%)	0.03(0.02, 0.05) ^b,c^	0.02 (0.01, 0.03) ^a,d^	0.01 (0.01, 0.03) ^a,d^	0.03 (0.01, 0.05) ^b,c^	0.000 **
Protein, E%	16.3(15.0, 17.6) ^b,c^	15.7 (14.0, 17.3) ^a,d^	15.4 (13.9, 16.3) ^a,d^	16.6 (15.0, 17.9) ^b,c^	0.000 **
Fat, E%	48.3 ± 5.6 ^b,c^	44.6 ± 5.6 ^a,d^	46.2 ± 5.8 ^a^	47.2 ± 6.2 ^b^	0.000 **
Carbohydrate, E%	35.3 ± 6.2 ^b,c^	39.7 ± 6.4 ^a,d^	38.5 ± 6.5 ^a,d^	36.3 ± 7.7 ^b,c^	0.000 **
**Other nutrients**					
Vitamin B_3_, mg/d	15.3 (11.0, 18.7) ^c^	13.8 (10.0, 18.7) ^c^	17.2 (13.8, 23.0) ^a,b^	14.8 (11.8, 21.4)	0.004 **
Vitamin C, mg/d	47.9 (36.8, 74.9) ^b,d^	84.7 (56.3, 104.8) ^a,c^	50.9 (34.9, 77.3) ^b,d^	74.3 (47.3, 100.2) ^a,c^	0.000 **
Potassium, mg/d	1930.9 (1449.9, 2539.2) ^b^	2510.9 (2024.8, 3142.3) ^a,c,d^	2072.5 (1682.4, 2706.3) ^b^	2038.5 (1487.8, 2871.1) ^b^	0.000 **
SFA, g/d	24.7(17.5, 32.6)	24.3 (19.2, 33.6) ^c^	26.4 (20.9, 37.1) ^b^	24.4 (18.5, 33.6)	0.152
MUFA, g/d	30.7(21.0, 42.7) ^b^	28.5 (21.6, 41.0) ^a^	33.9 (25.5, 47.0)	30.4 (21.9, 43.1)	0.021 *
PUFA, g/d	15.8(11.3, 24.6) ^c,d^	12.8 (9.1, 18.9) ^c,d^	14.8 (10.6, 24.2) ^a,b^	14.4 (11.0, 21.3) ^a,b^	0.019 *
DHA, g/d	0.001 (0.000, 0.002) ^b–d^	0.000 (0.000, 0.001) ^a^	0.000 (0.000, 0.001) ^a^	0.000 (0.000, 0.001) ^a^	0.000 **
EPA, g/d	0.002 (0.001, 0.012) ^b,c^	0.001 (0.001, 0.002) ^a,d^	0.001 (0, 0.013) ^a^	0.001 (0.001, 0.01) ^b^	0.000 **
**Dietary scores**					
**N**	122	122	122	122	
**DBI-16**					
HBS	12 (9, 16)	10 (7, 12)	13 (10, 16)	13 (9, 16)	0.000 **
LBS	15 (12, 17)	13 (11, 16)	16 (14, 20)	15 (12, 17)	0.000 **
DQD	26 (22, 32)	23 (20, 27)	30 (27, 34)	27 (23, 32)	0.000 **
**DII**	8.31 ± 5.19 ^b^	11.36 (9.25, 14.04) ^a^	10.07 (7.93, 13.32)	10.86 (6.33, 13.93)	0.000 **

Abbreviation: BMI: body mass index; BMR: basal metabolic rate; PBF: percentage body fat; FMI: fat mass index; VFA: visceral fat area; FFM: fat-free mass; FFMI: fat-free mass index; SFA: saturated fatty acid; MUFA: monounsaturated fatty acid; PUFA: polyunsaturated fatty acid; DHA: Docosahexaenoic acid; EPA: Eicosapentaenoic acid; DBI-16: diet balance index-16; HBS: high bound score; LBS: low bound score; DQD: diet quality distance; DII: dietary inflammatory index. * *p* < 0.05, ** *p* < 0.01. For the DBI-16, The closer the score is to 0, the closer the food intake is to the recommended intake. For the DII, higher scores are more pro-inflammatory and lower scores are anti-inflammatory. ^a^
*p* < 0.05, statistically different with pattern 1; ^b^
*p* < 0.05, statistically different with pattern 2; ^c^
*p* < 0.05, statistically different with pattern 3; ^d^
*p* < 0.05, statistically different with pattern 4. ^e^
*p* < 0.05, statistical difference between male and female.

**Table 4 nutrients-14-04544-t004:** The association of body composition, dietary scores, and dietary patterns with BMI.

Variable	Models	β (95% CI)	*p*
**Body composition**			
FMI	1	1.000 (0.998, 1.002)	0.000 **
	2	0.999 (0.997, 1.002)	0.000 **
	3	1.000 (0.997, 1.002)	0.000 **
FFMI	1	1.002 (0.999, 1.004)	0.000 **
	2	1.003 (0.999, 1.007)	0.000 **
	3	1.000 (0.994, 1.006)	0.000 **
**Dietary patterns**			
Pattern 1	1	−0.030 (−0.294, 0.235)	0.826
	2	−0.062 (−0.325, 0.200)	0.641
	3	−0.279 (−0.590, 0.032)	0.079
Pattern 2	1	0.246 (−0.023, 0.515)	0.073
	2	0.339 (0.068, 0.610)	0.014 *
	3	−0.032 (−0.478, 0.413)	0.886
Pattern 3	1	0.326 (0.056, 0.595)	0.018 *
	2	0.235 (−0.039, 0.508)	0.092
	3	−0.151 (−0.603, 0.301)	0.512
Pattern 4	1	0.160 (−0.105, 0.425)	0.237
	2	0.030 (−0.241, 0.302)	0.826
	3	−0.067 (−0.334, 0.201)	0.625
**Dietary scores**			
DBI-16			
HBS	1	NA	NA
	2	−0.044 (−0.096, 0.008)	0.095
	3	−0.059 (−0.102, 0.016)	0.007 **
LBS	1	0.076 (−0.012, 0.165)	0.091
	2	0.035 (−0.036, 0.105)	0.337
	3	0.046 (−0.012, 0.103)	0.117
DQD	1	−0.020 (−0.070, 0.031)	0.439
	2	NA	NA
	3	NA	NA
DII	1	0.065 (0.022, 0.107)	0.003 **
	2	0.046 (0.002, 0.089)	0.042 *
	3	NA	NA

Abbreviation: FMI: fat mass index; FFMI: fat-free mass index; DBI-16: diet balance index-16; HBS: high bound score; LBS: low bound score; DQD: diet quality distance; DII: dietary inflammatory index. NA, not available. * *p* < 0.05, ** *p* < 0.01. Model 1: unadjusted; Model 2: adjusted for age and gender; Model 3: additionally adjusted for BMR and energy based on model 2.

**Table 5 nutrients-14-04544-t005:** Factors contributing to different BMI.

Variable	Thinness		Overweight		Obesity	
	OR (95% CI)	*p*	OR (95% CI)	*p*	OR (95% CI)	*p*
**Body composition**						
FMI (per 1 kg/m^2^ increase)	0.07 (0.03, 0.18)	0.000 **	15.90 (7.76, 32.59)	0.000 **	NA	NA
FFMI (per 1 kg/m^2^ increase)	0.08 (0.03, 0.20)	0.000 **	5.91 (3.65, 9.58)	0.000 **	15.29 (6.92, 33.77)	0.000 **
Waist–hip ratio (Ref: male < 0.9; female < 0.85)	0.70 (0.04, 3.34)	0.717	0.25 (0.09, 0.67)	0.006 **	0.06 (0.00, 1.20)	0.065
VFA (Ref: < 100)	NA	NA	6.83 (2.54, 18.40)	0.000 **	20.61 (0.22, 1912.86)	0.191
PBF (Ref: male < 25%; female < 30%)	0.05 (0.01, 0.26)	0.000 **	18.69 (3.71, 94.25)	0.000 **	15.52 (0.04, 5417.42)	0.359
Abdominal obesity	NA	NA	9.77 (2.86, 33.34)	0.000 **	NA	NA
**Dietary pattern**						
Pattern 1						
Q1 (Ref)						
Q2	3.11 (0.66, 14.67)	0.151	1.13 (0.39, 3.29)	0.828	2.37 (0.28, 21.11)	0.431
Q3	4.04 (0.88, 18.57)	0.073	1.16 (0.41, 3.26)	0.776	0.62 (0.03, 11.12)	0.743
Q4	1.94 (0.34, 11.1)	0.456	2.21 (0.79, 6.14)	0.129	0.45 (0.04, 4.61)	0.498
Pattern 2						
Q1 (Ref)						
Q2	1.57 (0.46, 5.42)	0.474	1.63 (0.60, 4.38)	0.337	0.16 (0.01, 3.16)	0.229
Q3	0.24 (0.05, 1.13)	0.071	0.87 (0.28, 2.72)	0.807	0.72 (0.04, 12.44)	0.823
Q4	0.61 (0.08, 4.50)	0.630	1.77 (0.54, 5.79)	0.348	0.16 (0.01, 3.95)	0.266
Pattern 3						
Q1 (Ref)						
Q2	1.02 (0.26, 4.04)	0.980	1.27 (0.45, 3.56)	0.654	0.19 (0.01, 2.62)	0.214
Q3	1.72 (0.45, 6.64)	0.431	1.57 (0.52, 4.81)	0.427	0.19 (0.01, 3.97)	0.284
Q4	2.12 (0.39, 11.62)	0.387	1.33 (0.38, 4.62)	0.653	0.15 (0.01, 2.77)	0.201
Pattern 4						
Q1 (Ref)						
Q2	1.31 (0.31, 5.47)	0.716	1.35 (0.50, 3.68)	0.556	0.04 (0.00, 0.95)	0.047 *
Q3	1.21 (0.28, 5.12)	0.800	1.86 (0.66, 5.23)	0.237	0.75 (0.06, 9.59)	0.822
Q4	2.06 (0.45, 9.41)	0.350	1.37 (0.47, 3.97)	0.568	0.68 (0.08, 5.54)	0.719
**Dietary scores**						
DBI-16						
HBS						
Almost no problem (Ref)						
Low level	2.72 (0.89, 8.30)	0.079	1.19 (0.61, 2.32)	0.616	0.38 (0.11, 1.33)	0.130
Moderate and high level	3.07 (0.47, 19.98)	0.241	2.20 (0.59, 8.20)	0.241	0.17 (0.01, 2.40)	0.188
LBS						
Almost no problem (Ref)						
Low level	3.02 (0.68, 13.38)	0.145	1.10 (0.50, 2.42)	0.821	0.33 (0.08, 1.41)	0.135
Moderate and high level	26.62 (2.03, 349.71)	0.013 *	8.04 (1.28, 50.51)	0.026 *	1.90 (0.11, 33.24)	0.659
DQD						
Almost no problem (Ref)						
Low level	0.07 (0.01, 0.49)	0.008 **	0.73 (0.18, 3.02)	0.664	1.38 (0.16, 12.25)	0.774
Moderate and high level	0.03 (0.00, 0.40)	0.008 **	0.19 (0.03, 1.40)	0.104	3.72 (0.12, 111.20)	0.449
DII						
Q1 (Ref)						
Q2	1.54 (0.46, 5.13)	0.480	2.26 (0.96, 5.31)	0.062	1.25 (0.24, 6.61)	0.790
Q3	0.94 (0.19, 4.62)	0.943	1.00 (0.37, 2.73)	0.999	0.81 (0.13, 4.99)	0.817
Q4	0.73 (0.06, 8.41)	0.800	0.84 (0.25, 2.85)	0.781	0.44 (0.04, 4.81)	0.504

Abbreviation: FMI: fat mass index; FFMI: fat-free mass index; FFM, fat-free mass; VFA: visceral fat area; PBF: percentage body fat; DBI, diet balance index; HBS, high bound score; LBS, low bound score; DII, dietary inflammatory index; NA, not available. Ref, reference group in the regression. * *p* < 0.05, ** *p* < 0.01. Model is adjusted for gender, age, BMR, and energy.

**Table 6 nutrients-14-04544-t006:** The association of dietary scores and dietary patterns with FMI and FFMI.

Variable	Models	FMI		FFMI	
		β (95% CI)	*p*	β (95% CI)	*p*
**Dietary patterns**					
Pattern 1	1	−0.113 (−0.314, 0.089)	0.272	0.086 (−0.082, 0.253)	0.315
	2	−0.080 (−0.268, 0.109)	0.407	0.020 (−0.099, 0.139)	0.738
	3	−0.171 (−0.431, 0.089)	0.196	−0.104 (−0.211, 0.003)	0.057
Pattern 2	1	0.325 (0.120, 0.530)	0.002 **	−0.080 (−0.250, 0.090)	0.355
	2	0.165 (−0.029, 0.358)	0.095	0.185 (0.064, 0.306)	0.003 **
	3	0.000 (−0.372, 0.372)	0.998	−0.032 (−0.185, 0.122)	0.686
Pattern 3	1	0.006 (−0.199, 0.211)	0.953	0.318 (0.148, 0.488)	0.000 **
	2	0.201 (0.006, 0.395)	0.043 *	0.046 (−0.077, 0.168)	0.467
	3	0.033 (−0.344, 0.411)	0.863	−0.183 (−0.338, −0.027)	0.021 *
Pattern 4	1	−0.225 (−0.427, −0.023)	0.029 *	0.384 (0.217, 0.552)	0.000 **
	2	−0.002 (−0.196, 0.192)	0.985	0.03 (−0.092, 0.152)	0.627
	3	−0.038 (−0.261, 0.186)	0.741	−0.028 (−0.121, 0.064)	0.545
**Dietary scores**					
DBI-16					
HBS	1	−0.074 (−0.110, −0.038)	0.000 **	0.036 (0.005, 0.067)	0.022 *
	2	−0.040 (−0.074, −0.005)	0.024 *	−0.021 (−0.043, 0.001)	0.058
	3	−0.036 (−0.069, −0.004)	0.029 *	−0.018 (−0.031, −0.004)	0.010 **
LBS	1	−0.010 (−0.056, −0.036)	0.680	0.003 (−0.036, 0.042)	0.874
	2	0.013 (−0.030, 0.056)	0.550	−0.031 (−0.058, −0.005)	0.022 *
	3	0.037 (−0.005, 0.079)	0.082	−0.009 (−0.026, 0.009)	0.330
DQD	1	−0.041 (−0.066, −0.015)	0.002 **	0.019 (−0.003, 0.041)	0.087
	2	−0.016 (−0.04, 0.009)	0.209	−0.021 (−0.036, −0.006)	0.007 **
	3	−0.007 (−0.031, 0.017)	0.553	−0.012 (−0.022, 0.002)	0.015 *
DII	1	0.011 (−0.016, 0.038)	0.417	0.041 (0.018, 0.063)	0.000 **
	2	0.024 (−0.011, 0.037)	0.061	0.022 (0.007, 0.038)	0.005 **
	3	NA	NA	NA	NA

Abbreviation: FMI: fat mass index; FFMI: fat-free mass index; DBI, diet balance index; HBS, high bound score; LBS, low bound score; DII, dietary inflammatory index; NA, not available. * *p* < 0.05, ** *p* < 0.01. Model 1: unadjusted; Model 2: adjusted for age, and gender; Model 3: additionally adjusted for BMR and energy based on Model 2.

## Data Availability

Data are available upon reasonable requests.

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
