# Peer review of "The Role of Dietary Patterns and Dietary Quality on Body Composition of Adolescents in Chinese College"

_nutrients, 2022, doi:10.3390/nu14214544_

Round 1

Reviewer 1 Report

Thank you for submitting the manuscript “The Role of Dietary Patterns and Dietary Quality on Body Composition of Adolescence in Chinese College” to Nutrients.

The subject of the work is interesting, but I have some concerns about the methodology. If the sampling was not carried out randomly, it will be impossible to say that a pattern has been established (as indicated in the title and in the objectives). This sampling needs to be better described in the M&M. Also, I suggest some other small corrections.

Line#16: was

Line#21: add definition for BMI, FMI, and FFMI

Line#22: add definition for “ORs”

Line#29: add definition for “VFA”

Line#30: add definition for “HBS”

Line#32: add definition for “DQD”

It is practically impossible to understand the abstract due to the large number of abbreviations that the text contains. Authors need to remember that the reader of Nutrients will often read the abstract first and, if interested, the full article. My suggestion is to reformulate, making it lighter and clearer only with the main results.

My suggestion is to change some keywords to keywords that are not in the title.

Line#53: add definition for “BMR”

Line#58: add definition for “DASH”

Line#60: I don't believe readers can know what cardiorespiratory fitness is.

Line#77: Here in this line, after being talked about BMI several times, the authors decided to define the abbreviation.

Line#92: How were these students selected? Was more than one school used? The objective of the work is to find a pattern and this should be based on good sampling.

Line#186: This is interesting information that is different from what the WHO has been releasing.

Lines#345-346: This doesn't seem new to me.

Line#424: correct this citation

Line#444: is this really a concern? This value is much lower than the value found for western countries.

Reviewer 2 Report

In this cross-sectional study, the authors explored the role of dietary patterns and dietary quality on body composition and BMI of youth in Chinese college participants. Body composition assessment is an important challenge nowadays and although the topic is generally worthy of consideration, my main concern is about the BIA data and their collection. Detailed comments are listed below. 

Introduction

- From line 70 to line 78 the introduction fails to provide an overview regarding body composition assessment techniques. Key studies from the past 10 years are missing and my suggestion is to check the MDPI literature (see: Nutrients 2021, Assessment of Body Composition in Athletes: A Narrative Review of Available Methods with Special Reference to Quantitative and Qualitative Bioimpedance Analysis.) I would encourage to authors to read more around the subject as these aspects need to be considered in the introduction to form a basis for this paper. 

Methods

- Please provide results for a power analysis that indicate your sample size was appropriate.

- 2.5. section

- Anthropometric measurements should be changed in “body composition assessment”

- There is a basic need to describe the technical characteristics of the Inbody device. What is the calibration method to ensure validity (accuracy and precision) of the bioimpedance measurements? What is the technical error of measurement in vivo? Provide readers with a concise description of what this BIA device measures. In particular, what are the measurements detected by this tool? Do they directly measure the raw bioimpedance parameters (e.g., R, Xc and phase angle)? Again, what equation was used to estimate body composition indexes? Are them equations developed using the Tanita device or an instrument that works with similar characteristics (frequency and technologies)? 

Results

- Using BIA data without considering the raw parameters (e.g., resistance, reactance, and phase angle) especially in the vector approach (BIVA) is a poor way to use BIA data. The vector analysis (BIVA) is also useful in comparing body composition features with the reference population. However, my concern is about the lack in literature of specific reference provided using the direct segmental BIA (standing position)

Discussion

- The discussion section is very descriptive and offers limited comparisons to previous research. How do practitioner benefit from that? Again, the discussion section fails to relate the findings to this particular application of interest. Moreover, it is important to consider that BIA data are dependent instrument and that the instrumental sensitivities are different. Therefore, no comparisons can be made between studies that measure phase angle with different technologies (e.g., foot-to-hand- or direct segmental in standing position) or sampling frequencies. Authors are therefore encouraged to make substantial changes throughout to improve the overall quality. In the current form the rationale for the study is not clear, the new value is unclear, and I have difficulties finding specific take home messages for practitioners.

Round 2

Reviewer 2 Report

Authors addressed all my comments